# *Treponema denticola* Has the Potential to Cause Neurodegeneration in the Midbrain via the Periodontal Route of Infection—Narrative Review

**DOI:** 10.3390/ijerph20116049

**Published:** 2023-06-04

**Authors:** Flavio Pisani, Valerio Pisani, Francesca Arcangeli, Alice Harding, Simarjit Kaur Singhrao

**Affiliations:** 1Faculty of Clinical and Biomedical Sciences, School of Dentistry, University of Central Lancashire, Preston PR1 2HE, UK; 2IRCCS, “Santa Lucia” Foundation, Neurology and Neurorehabilitation Unit, Via Ardeatina, 306, 00179 Rome, Italy; 3Azienda Sanitaria Locale ASLRM1, Nuovo Regina Margherita Hospital, Geriatric Department, Advanced Centre for Dementia and Cognitive Disorders, Via Emilio Morosini, 30, 00153 Rome, Italy; 4Dementia and Neurodegenerative Disease Research Group, Faculty of Clinical and Biomedical Sciences, School of Dentistry, University of Central Lancashire, Preston PR1 2HE, UK

**Keywords:** Alzheimer’s disease, axonal transport, cytoskeletal impairment, mitochondrial docking, neurodegeneration, peripheral nerve, periodontal disease, *Treponema denticola*

## Abstract

Alzheimer’s disease (AD) is a neurodegenerative disease and the most common example of dementia. The neuropathological features of AD are the abnormal deposition of extracellular amyloid-β (Aβ) and intraneuronal neurofibrillary tangles with hyperphosphorylated tau protein. It is recognized that AD starts in the frontal cerebral cortex, and then it progresses to the entorhinal cortex, the hippocampus, and the rest of the brain. However, some studies on animals suggest that AD could also progress in the reverse order starting from the midbrain and then spreading to the frontal cortex. Spirochetes are neurotrophic: From a peripheral route of infection, they can reach the brain via the midbrain. Their direct and indirect effect via the interaction of their virulence factors and the microglia potentially leads to the host peripheral nerve, the midbrain (especially the locus coeruleus), and cortical damage. On this basis, this review aims to discuss the hypothesis of the ability of *Treponema denticola* to damage the peripheral axons in the periodontal ligament, to evade the complemental pathway and microglial immune response, to determine the cytoskeletal impairment and therefore causing the axonal transport disruption, an altered mitochondrial migration and the consequent neuronal apoptosis. Further insights about the central neurodegeneration mechanism and *Treponema denticola*’s resistance to the immune response when aggregated in biofilm and its quorum sensing are suggested as a pathogenetic model for the advanced stages of AD.

## 1. Introduction

Research-based evidence continues to strengthen how periodontitis and Alzheimer’s disease (AD) may be comorbid. Moreover, it has also been reported that neurodegenerative diseases, such as anxiety disorders, depression, bipolar disease, and schizophrenia, are also associated with periodontal disease [1]. Cross-sectional studies have demonstrated a high prevalence of tooth loss and increased prevalence of periodontal disease in AD patients [2,3], but whether this is a downstream effect of factors associated with poor oral hygiene and periodontitis or simply a bidirectional relationship in dementia patients is yet to be clarified [4,5,6,7,8]. Defining relationships proves difficult because there appear to be multiple pathways to AD initiation, including genetics, traumatic brain injury, comorbidities, pollution, and lifestyles, along with many others [9,10,11].

AD is characterised by its two neuropathological hallmarks—amyloid-beta (Aβ) plaques and neurofibrillary tangles (NFTs) in the cerebral cortex progressing to the entorhinal cortex medial temporal lobes, including the hippocampus. According to the severity as defined by Braak and Braak [12], AD progression occurs throughout the cerebral cortex and the midbrain where the locus coeruleus (LC) is located. Nevertheless, the hippocampus bears the major impact of the disease process due to the major neuropathological alterations described within this anatomical site of the AD brain [12,13]. In light of the hypothesis that AD may initiate from the midbrain and spread to the cortical brain, the present narrative review aimed to focus on the role and involvement the LC and the brainstem specifically play in the pathogenesis of early AD [14,15,16,17]. NFTs are one of the two lesions that characterise AD [18], which correlate with clinical symptom onset during the Braak stages I to IV [19]. Several recent studies have shown that the LC develops phosphorylated tau (p-tau) early on, as acknowledged in the amended neuropathological staging of AD by including a stage 0 to precede stages I to IV [13]. This concept questions whether disease progression should be related to NFT spread within the brain.

The basis for supporting the hypothesis of NFT spread irrespective of where that starts within the brain, the present article relates to LC degeneration in the context of impaired noradrenaline (NE) neurotransmitter modulation [20]. Insufficient NE has an impact on blood pressure and would likely demonstrate a negative impact on cerebral tight junction proteins and subsequently on blood–brain barrier (BBB) permeability [21]. The subtle structural changes in BBB integrity will give rise to neuroinflammation (microglial activation) [22,23,24] and oxidative stress [25,26], all of which are attributable to infection since the LC lacks Aβ deposition at the 0 neuropathological staging of AD [13,27].

Regarding the close proximity of the LC to the trigeminal mesencephalic nucleus (VMes) [28,29], research suggests that this is where all the proprioceptive neural pathways from the peripheral nerve endings would project to if located in the periodontal ligament and in the surrounding dental tissues [30,31,32,33]. In line with this association, it is plausible to suggest that even an inflammatory insult due to traumatic injury, such as a complete deafferentation due to tooth extraction [34,35], could reasonably cause irreversible nerve damage in the subcortical nuclei with the consequent risk to the neuronal cell bodies resulting in their apoptosis, thereby leading to NE deficit [18,36]. Clinically, this may manifest as an impairment in masticatory function, which in turn is associated with symptoms akin to mild cognitive impairment in would-be AD subjects [28,37,38]. Alternatively, a peripheral infective assault from pathogenic oral bacteria could initiate neuronal axonal and/or dendritic degeneration from within the oral tissues, as the impact of peripheral inflammation encourages vascular leakage and bacterial entry into the bloodstream [39,40]. 

A symbiotic relationship links *Treponema denticola* and *Porphyromonas gingivalis*, both of which belong to the red complex consortium of bacteria involved in periodontal infections, whereby *T. denticola* would achieve its motility from *P. gingivalis* [41], while sustenance and growth of *P. gingivalis* would be supported via the spirochete triggering physical damage of tissue [42].

Several studies investigating the impact of treponemes on neuronal cells have demonstrated that bacterial-associated cytotoxic effects lead to an initial electrophysiological impairment of neurons and later to apoptosis [43]. The present narrative review specifically discusses some of the mechanisms employed by *T. denticola* and other treponemes that may potentially contribute to LC neurodegeneration and NE deficit.

## 2. *T. denticola* and Its Virulence Factors

Treponemes are etiological factors of different human chronic diseases, such as syphilis (*Treponema pallidum*), acute necrotizing ulcerative gingivitis or periodontitis (*T. denticola*, *T. lecithinolyticum*, *T. socranskii*), and endodontic infections [44,45,46]. They belong to the Spirochaetes taxonomic phylum, which differs from the common Gram-negative and Gram-positive bacterial phyla. Spirochetes show some unique features, such as specific virulence determinants in their metabolic pathways. They have a unique solute transport system, which is located within their outer surface membrane binding proteins [47,48].

Within the oral biofilm, they can be part of the normal healthy microflora if in fewer numbers. However, they can increase in number and interact with *P. gingivalis* and *Tannerella forsythia* (also a member of the red consortium of bacterial complexes) to become the main etiological agents in periodontitis and in most of the acute necrotizing gingival and periodontal diseases [45,49,50]. Together these micro-organisms cause tissue damage via multiple virulence factors, including proteolytic exotoxin activity, the complex anaerobic fermentation of some amino acids, and the production of toxic metabolites and outer membrane vesicles [51,52]. 

The major encounterable *T. denticola* virulence factors exist within the toxin–antitoxin system (TA) [53,54], namely Transposases [55]. These include the outer sheath proteins, the major sheath protein (Msp) [56,57], trypsin-like protease activity [58,59,60], lipoproteins [61], the outer sheath vesicles [62,63,64], and dentilisin [65,66,67,68,69]. The TA system consists of both a toxin, which inhibits essential cell components, and an antitoxin, which counteracts the toxic effects on the self. The virulence factors are involved in several roles, such as programmed cell death, the response to amino acid starvation, and the reversible effect of bacteriostasis. These roles may influence biofilm formation [53,54] and represent a resistance mechanism to environmental attacks, antibiotics, and other drugs [70,71]. *T. denticola* transposases are enzymes that “cut and paste” mobile genetic elements from one position to another within the host genome. This effect may lead to chromosomal re-arrangement or may represent a novel gene regulatory mechanism [55].

A major feature common to the bacteria belonging to the red complex is their high levels of extracellular proteolytic activity located on the outer cell surface. At the same time, these external components may represent the sensors and effectors within the host adaptive immune response along with other bacteria within the biofilm consortium. Trypsin-like protease activity, reportedly from *T. denticola* [58], is related to an oligopeptidase that has been shown to cleave only C-terminal Arg-residues [59,60], and to another protease, opdB, that may have a more Lys- specific action. 

*T. pallidum* Msp, which is part of the *T. pallidum* repeat (Tpr) protein family, is the most abundant protein in the *T. denticola* outer membrane. It has a β-barrel tertiary structure contained within its integral outer membrane protein that acts as a porin, which can bind to different host proteins. Msp mediates the colonization of *T. denticola* to host tissue whilst protecting itself from the cytopathic pore-forming activity against epithelial cells [56,57]. Msp is also able to bind keratin, collagen type 1, fibrinogen, hyaluronic acid, and heparin via its major epitopes located in the N-terminal domain [72]. Lipoproteins are the most abundant membrane-associated proteins in spirochetes, up to 166 in *T. denticola* alone [61]. Of these, OppA can bind soluble host proteins, such as plasminogen and fibrinogen, but not immobilized insoluble host proteins or epithelial cells. It has been proposed that OppA may act as an adhesin and aid *T. denticola* with the evasion of host-mediated immune recognition [73]. 

Gram-negative bacteria have long been known to produce outer membrane vesicles (OMVs) [74,75]. Initially, OMVs were thought to result from random blebbing of the outer membrane with small spherical vesicles of 50–100 nm in diameter. However, current research has confirmed OMVs are an adaptation of the bacterium to its local environment [62,63,64]. OMVs are potent virulence factors since they possess adhesins, toxins, and proteolytic enzymes that can mediate bacterial aggregation and invasion, host immune response modulation, and tissue invasion [76,77,78] via disruption of tight junctions [79] and can impart an effective cytotoxic effect on the host [78].

Dentilisin is another tool in the armamentarium of *T. denticola*’s major virulence components. Dentilisin is an active cell surface-located protease with the ability to cleave phenylalanyl/alanyl and prolyl/alanyl bonds [65,67,68,69,75]. It is effective both in the combined disruption of the intercellular host signalling pathways along with the degradation of host cell matrix proteins, intercellular adhesion proteins, and tight junction proteins, such as Zonula Occludens 1 (ZO-1), thereby increasing BBB permeability [79]. It has the ability to degrade interleukin-1β (IL-1β), IL-6, tumour necrosis factor alpha (TNF-α), and monocyte chemoattractant protein 1 [80,81]. Conversely, it induces matrix metalloproteinase-2 activation via the fragmentation of fibronectin within the extracellular matrix [82], thus further enhancing tissue invasion and destruction.

## 3. The Mechanism of Peripheral Neurodegeneration via *T. denticola*

### 3.1. The Role of *T. denticola* in Peripheral Nerve Damage

Laboratory investigations based on in vitro cultured cells have demonstrated a close interaction between *T. pallidum* and nerve cells derived from rat embryogenic dorsal root ganglia [83]. Here, treponemes were shown to be attached to neuronal and fibroblastic cells, retaining the typical motility of bending, flexing, and rapidly rotating about their axis but without moving from their site of adhesion to the cell. The attachment was commonly mediated through the tip of one end of the treponemes’ body, while only a few cells displayed a double attachment, keeping only the middle portion detached from the cell enabling rotation. Besides attaching to the nerve cell body, *T. pallidum* was also attached to the neuritic processes that extended outwards from the nerve cells. Another in vitro study that utilised rat embryonic nerve cells seeded at a density of 2 × 10^8^/mL demonstrated that *T. pallidum* could cause electrophysiological impairment after 18 h and a substantial morphological alteration after 16 h exposure to the cell. The nucleus of the infected neurons appeared atypical, and the overall shape of the cells changed from their original rounded shape to a flattened one. In addition, cell death resulted from pores created in the surface membrane of eukaryotic cells [43]. This may explain one mechanism as to how the NE-synthesising neurons die off, creating a neurotransmitter deficit within the LC.

An electron microscopy observation on human-derived syphilitic chancres has shown spirochaetes gathering around peripheral nerves and invading the spaces between Schwann cells and their basal lamina either within myelinated or unmyelinated fibres. In some cases, treponemes were engulfed by the perineural cells but showed no evidence of destruction or cell degeneration [84]. Degenerated axons of syphilitic chancres were demonstrated by Wrzolkowa and Kozakiewics [85], also suggesting the involvement of nerves, but in this case, no treponemes were identified. In experimental syphilomas of rabbits, Ovcinnikov and Delektorskij [86,87] identified a small number of *T. pallidum* cells between the collagen fibrils of the endoneurium, but again, there was no evidence of selective bunching (mini biofilms) typical of treponemes within the basal lamina of Schwann cells.

### 3.2. The Evasion of the Complement Cascade

The complement system is part of the body’s innate immune defence mechanism. One of the many functions of complement activation in health is to facilitate the clearance of foreign agents in relation to bacterial infections by coating microbes with immune complexes and opsonins (C1q, C3b and iC3b) via the activation of one of the three (classical, alternative, lectin) possible pathways, all merging at the C3 convertase stage. Stimulation of glial cells by complement activation products (C3a/C5a) gives rise to inflammation and the release of inflammatory mediators [23,88,89,90]. This leads to chronic neuroinflammation and increased levels of cytokines and is accompanied by synapse loss. These features overlap with AD pathophysiology [91,92]. One of the main pathways leading to neuronal damage and synapse disruption is via classical complement pathway (CP) activation by bacteria and inadvertently by gene defects. There is evidence supporting that the CP is activated in AD patients’ brains [22,93,94,95].

The CP in the brain can initiate the binding of C1q to apoptotic cells, pathogens, and malfunctioning synapses, leading to their demise with C3b fragment opsonization. This in turn triggers microglia to phagocytose the opsonized connections [96]. Central astrocytes are involved in the complement-dependent removal of excitatory and inhibitory synapses [97], showing cooperation with microglial cells over time [98]. Studies on gene knock-out mice for complement components have demonstrated glial cells engulfing functional synapses either by over-pruning mechanisms or clearing of complement-labelled debris [94,99].

Factor H (FH) is a fluid phase complement regulatory protein of the alternative complement activation pathway. Previous studies have reported that *T. denticola* binds at least one member of the FH family proteins [100], which include FH, FH-like protein 1 (FHL-1), and five other FH-related proteins [101]. FH and FHL-1 serve as cofactors in the factor I-mediated cleavage of C3b and accelerate the decay of the C3 convertase complex, leading to the downregulation of C3b production [102,103]. Evasion of complement by *T. denticola* via FhbB binding is further enhanced by the cleavage of FH due to dentilisin, thereby leading to a local dysregulation of complement activation with severe tissue destruction and bone resorption, as is seen in periodontal disease (Figure 1) [104].

Another mechanism enabling *T. denticola* to evade destruction via complement is linked with its own neuraminidase TDE0471. This enzyme is a cell surface-exposed exo-neuraminidase that removes sialic acid from human serum proteins, suggesting that sialic acid can act as a main carbon source of nutrients for *T. denticola* growth and colonisation. This neuraminidase also protects oral spirochaetes from destruction by the serum by preventing the deposition of membrane attack complexes on their cell surface, but paradoxically, the lack of a full bacterial effect shifts all the inflammatory damage onto the periodontal tissue and further bystander damage to neuronal cells (Figure 1) [105].

### 3.3. Cytoskeletal Impairment

Host nerve cells are also under attack from its related signalling pathways, which are activated as a result of bacterial infection [106]. This mainly occurs due to virulence factors interacting directly with host cell cytoskeletal components or cytoskeletal regulatory proteins upon infection or to the modulation of signal transduction pathways (small GTPases) as the actin remodels [107,108]. *T. denticola* is known to disrupt the actin cytoskeleton of fibroblasts, epithelial cells, and neutrophils [106,109,110]. Msp perturbs cell actin dynamics, leading to impaired cell migration and chemotaxis [111,112,113]. In particular, it impairs calcium signalling and collagen binding within cells, as well as inciting cell shrinkage and rounding up, most likely due to Msp-induced subcortical actin assembly near the plasma membrane [114,115]. 

The actin-binding protein cofilin serves two functions within the cell: firstly, to depolymerize the actin filaments to promote its turnover and secondly to sever filaments to create free barbed ends (FBEs). Cofilin usually remains inactive in resting cells, as it is inhibited either by binding to Phosphatidylinositol 4,5 biphosphate (PIP2) [116] or by its phosphorylation at serine 3 [117]. Msp has the potential to directly perturb the cellular actin cytoskeleton [115,118]: Therefore, extracellular contact with *T. denticola* may induce phosphatase-enhancing PIP2 production, thus leading to cofilin inactivation and actin filament uncapping (Figure 1).

The unique morphology of neurons with extended axons and dendrites makes them rely on the active intracellular transport of proteins, RNA, and organelles over long distances with molecular motors carrying them along the cellular cytoskeleton. Two major roles are flagged for axonal transport: supply and/or clearance of energy and waste and a long-distance signalling process. Energy provision involves delivering proteins, lipids, and mitochondria to support synaptic function, while waste clearance involves the transport of misfolded and aggregated proteins to the central soma to be fully degraded [119]. 

The second major role of the active transport system is the communication of intracellular signals from the distal axon to the soma in order to respond to environmental changes. While the alteration of energy provision and clearance may be assumed to be deleterious for the neurons, growing evidence for altered signalling to be a key neurodegenerative pathway leading to cell death is under review [120,121].

The active transport and delivery of proteins and lipids along the axons to synapses represent a process called fast axonal transport (FAT) which is crucial for the neuron to function [122]. Conventional kinesins work as molecular motors within the brain [123], and they are involved in the anterograde FAT of membrane-bound organelles (MBOs) consisting of mitochondria, synaptic vesicles, and axolemmal precursors [124,125].

Conversely, cytoplasmic dynein is the major motor of retrograde axonal transport [126], and its impairment appears to lead to retrograde axonal transport dysfunction with altered transport of misfolded proteins and their degradation in the cell body. Furthermore, dynein is the molecular motor responsible for retrograde signalling from the synapse to the soma. This mechanism is involved in cell-to-cell communication within the nervous system, and its dysfunction might account for neuronal apoptosis in neurodegeneration. Even if this transport impairment reflects a direct inhibition of motor function, it may be possible that the defect arises from the dysregulation of axonal trafficking, which means an alteration in the microtubule track in cargo-specific adaptors or in scaffolding proteins coordinating cargo-bound motors [122]. The alteration of the cytoskeletal track may involve the acetylation [127] or tyrosination [128] of tubulin subunits or the modifications of MAPs (microtubule-associated proteins), which can compete with motors for binding to the microtubule surface [129,130,131], as it is observed in the case of tau protein and NFTs in AD. 

Mitochondria play essential functions in the maintenance of the viability of neurons, including oxidative phosphorylation to supply ATP, along with calcium homeostasis [132]. It has been shown that the interruption of mitochondrial function in neurons is typically associated with the initiation or the amplification of a neuronal injury [133,134,135]. While mitochondrial dynamics in neurons are straightforward, the purpose of mitochondrial movement (trafficking) or alterations in their morphology is less obvious. The physiological pattern would involve the generation of new mitochondria in the proximity of the neuronal nucleus with subsequent movement to sites of high ATP demand or calcium influx. Once their use has been exceeded, the trafficking of old and dysfunctional molecules to a subcellular graveyard ensues for their autophagic destruction [136]. A variety of neurotoxic stimuli may arrest mitochondrial movement and their fragmentation after neuronal injury. From the microscopic observations of cultured neurons, it is evident that mitochondria could move from one cell to another to provide support; however, some practical limitations of imaging preclude the exact tracking of such individual organelles over a relatively short distance.

The primary mechanism for mitochondrial movement requires support from microtubules. It is well documented that fast anterograde axonal movement is generated by plus end-directed kinesins, such as kinesin 1 and 3 [137]. Slow retrograde axonal movement is likely to be mediated by dynein proteins. Their cargo specificity is provided by a range of accessory proteins. However, mitochondria can also move along the actin cytoskeleton. Several forms of myosin are found within neuronal processes, and myosin V may be the most likely form to be involved in mitochondrial attachment and movement along the actin fibres [137].

There is growing evidence that proteins associated with neurodegenerative disease could impact the axonal transport of cellular organelles, especially mitochondria, forming aggregates in the dendrites and blocking their transportation (Figure 1) [136]. 

There are several reports describing other types of interactions with mitochondrial trafficking, including decreased mitochondrial velocity [138], or impairment of the charged mitochondria on microtubule-based transport proteins [139]. Microtubule-associated protein tau when hyperphosphorylated forms NFTs in AD [140]. Tau interferes with the attachment of cargoes to kinesin-based motors so that tau overexpression results in the accumulation of mitochondria near the minus end of microtubules within the cell centre [141]. In primary neurons, p-tau overexpression causes the depletion of mitochondria from dendrites and axons [142].

In general, the hyperphosphorylation of tau, as is typical of neurodegenerative diseases, would result in its dissociation from microtubules and their subsequent destabilization [143], which would interrupt the delivery of mitochondria to key cellular sites where there would be a greater demand for energy, for example, due to an injury.

In addition, amyloid precursor protein (APP) and apolipoprotein E4 allele inheritance can interact with mitochondria and impair their bioenergetic function [144,145] and mitochondrial trafficking. In AD, it is challenging to attribute the key effect to a single pathogenic mechanism. However, impairing the delivery of mitochondria to presynaptic sites in axons and the subsequent loss of synaptic activity provides an appealing justification to account for its progressive loss of synaptic connections. 

## 4. The Role of *Treponema denticola* in Central Neurodegeneration

The infectious hypothesis of AD was proposed when the evidence of spirochetes was detected in the brains of affected patients [146,147]. When vascular leakage occurs in peripheral tissues, the microbes could potentially be transported from the disrupted dental plaque into the bloodstream, thereby reaching different organs and supporting systemic inflammation. Alternatively, spirochetes have an affinity for neural tissue and in this way can avoid the BBB easily [148]. Once within the brain, they proliferate to form mini biofilms [149]. Quorum sensing (QS) enables the bacteria residing in biofilms to chemically communicate among themselves in order to respond to environmental changes by coordinating their activity as if they were multicellular organisms [150]. Bacteria can synthesize and export signalling molecules called autoinducers (Ais) [151,152], which, at a particular threshold, are able to change gene expression and therefore produce various virulence factors [153]. Once the biofilm has been established, the innate immune response system attempts a reaction against treponemes via toll-like receptor (TLR)-2 [154] instead of TLR-4. TLR-2 attacks the curli fibres (bacterial amyloids) within the biofilm, as they represent an attachment and a strong immunogenic factor shared by many Gram-negative bacteria, including those belonging to the *Enterobacteriaceae* family of bacteria [155]. Usually, TLR-2 coats the bacteria and activates nuclear factor-B (NF-kB) signalling to generate cytokines, such as TNF-α, to destroy them. However, this process can only work when treponemes are in their planktonic state, not when they are aggregated into mini biofilms; therefore, the non-specific activity of the innate immune system causes bystander damage to the surrounding neural tissue and causes indirect damage to the vulnerable neurons [154]. 

In more advanced stages of the neurodegenerative disease, BBB permeability increases further, and the adaptive immune system, including B cells, immunoglobulins, and T cells with their cytokines, can enter the brain and impart their destructive responses, especially around the Aβ plaques. In this regard, the role of Aβ, even if recent evidence has shown its antimicrobial peptide function, appears to protect and shield the biofilm enabling its continuity.

Human Aβ can be derived from APP via the TNF-α converting enzyme (TACE), acting as an alpha-secretase, that cleaves it as the soluble truncated alpha APP fragment. Together with the amyloid converting enzyme (BACE), the beta and gamma secretases are activated, and these cleave the APP protein to release Aβ. In its antimicrobial role, Aβ attacks the treponeme mini biofilms. Whether this leads to the effective destruction of the spirochete or to disruption of the bacterial mini biofilms remains unclear. It is more likely that dissociation or disruption of the mini spirochete biofilm is taking place, and this has negative connotations in that the individual spirochetes are then able to multiply to form more mini biofilms.

## 5. Discussion

We initially hypothesized that spirochetal infections of the AD brain may be able to trace AD pathology from the brainstem aspects [156]. A follow-up publication discussed how periodontal disease bacteria (spirochetes) could potentially damage LC neurons [157]. The present review delves more deeply into the possible bacterial mechanisms that could damage the NE neurons and lead to a neurotransmitter deficit. The infectious pathogenesis hypothesis of AD has become more popular due to the finding that Aβ is the host’s innate immune response to the microbial assault. After all, microbes bring an armamentarium of weapons to inflict immune injury on the host as highlighted by the aspect of spirochetes in the present review.

The evidence of treponemes with their entry in the ganglia from peripheral sites to the midbrain in autopsies of AD patients clearly shows these bacteria to be able to enter from the back of the brain and thus could develop pathology in the LC [147]. From here, it could potentially spread in a reverse manner to the frontal cortices by spreading along neural pathways associated with the ganglion roots.

A previous review from the same group [156] has shed some light on the potential for *T. denticola* to evoke peripheral neuronal damage rather than central neuronal invasion. The treponemes are more likely to be able to disseminate along the neural sheath rather than locate in the axons and dendrites.

The periodontal proprioceptors are part of a direct pathway to the subcortical nuclei (VMes) without any stop-over at the Gasserian ganglion. These nuclei show a critical proximity within the pons to the LC, which has been identified as being one of the earliest regions of the brain where neurodegeneration may start in AD even before the onset of signs and symptoms of mild cognitive impairment.

From here, LC degeneration would cause critical NE levels to decrease within the entire brain, supporting the diminished activity of the central nervous system. Rather than a true treponeme invasion of the Vmes and the LC due to their retrograde transport along the neurons, it is plausible to say that they may cause peripheral neuropathy, leading to apoptosis of the central neurons that synapse onto them.

Studies on peripheral deafferentation or tooth extraction in mice have shown how neurodegeneration of VMes neurons and the LC have occurred within five days without any attempts of recovery by the host as per Wallerian degeneration [18,158]. 

How *T. denticola* could be involved in peripheral neuronal damage along the periodontal tissues remains unexplored. This review supports the theory of oral treponemes to induce neurotoxic damage when deeply infiltrating the periodontal tissues in active periodontitis. Oral treponemes are able to evade the immune system and resist the activated complement cascade and microglial phagocytosis and can generate sialic acid as its nutritional source partly due to FhbB binding. *T. denticola* could use its virulence factors, Msp, and dentilisin to initiate neuronal cytoskeletal impairment, perturbing the actin cytoskeleton without invading or injecting any secreting effector within the cell [115]. The extracellular contact with *T. denticola* alone would be sufficient to induce phosphatase-enhancing PIP2 production leading and therefore to cofilin inactivation and actin filament uncapping.

The axonal transport systems, either the anterograde but mainly the retrograde mechanisms involving dyneins and myosins, would subsequently be impaired with the abnormal phosphorylated tau protein interacting with the microtubules, thus causing further trafficking in the neuron.

As the main result of cytoskeletal dysregulation, mitochondrial support to synaptic function and energy needs for damaged neuronal cells would be unmet due to their culling, and that would lead to programmed cell death in the Vmes.

As a consequence of further inflammatory damage and vascular impairment within the peripheral tissues, the ensuing leakage encourages entry of *T. denticola* and *P. gingivalis* into the blood, which would be responsible for their central dissemination. Once entry into the brain has been accomplished, the opportunity for *T. denticola* to build up as mini biofilms could protect it from being attacked by the innate immune system (TLR-2) initially and later offer protection from the adaptive response (lymphocytes B, IgG and TNF α). As *T. denticola* is part of the red complex consortium of bacteria, and as an individual species, it is inflammophilic and can tolerate high inflammatory milieus within the surrounding neural tissues, as is seen in AD. 

## 6. Conclusions

Recent research-based evidence has shown how periodontitis and neurodegenerative diseases may be comorbid and how they could potentially share the same bacterial consortium (*T. denticola* and *P. gingivalis*). This is possible due to their marked neurotropism and their neuroinflammatory effects along defined anatomical pathways, together with their ability to endure and flourish under inflammatory milieus influenced by the keystone periodontal pathogen *P. gingivalis*.

The bacterial hypothesis, even if identified as consistent based on brain autopsies and laboratory findings, is still lacking an accurate overview of the early stages of the AD disease process; hence, this initial peripheral damage, which this review focused on, would appear to represent a neurotoxic mechanism for inflicting and entering the brain, as shown with *T. denticola*, with the consequent neuronal damage via apoptosis rather than cell invasion along the midbrain pathways.

The evidence from cultured neurons with exposure to treponemes has demonstrated a functional impairment of their electrophysiological properties within 12 h without any evidence of bacterial intracellular localisation. Therefore, they would not appear to be intracellular bacteria.

The peripheral damage may lead to apoptotic death of neurons, encouraging the degeneration and consequent neuroinflammation to spread from the trigeminal nuclei to the limbic system via the LC and subsequently spread to the entire brain.

Interestingly, *T. denticola* forms mini biofilm aggregations within the central nervous system, and this acts to increase their effective virulence via quorum sensing, and the amyloid acting to disperse the biofilm encourages further growth and mini biofilm formation.

Further research is required to investigate the potential value of preventative use of the combination of repeated targeted antibiotics together with biofilm dispersers. Many AD patients are already on biofilm dispersers, such as donepezil, a chemical piperidine, or citalopram, a chemical furan, or haloperidol [154]. Investigations would need to assess the frequency of the antimicrobial treatment cycles and their duration, as encouraging results have been provided by some clinical trials. If this is the case, it would be important to define the most efficient antimicrobial agent, considering treponeme resistance to macrolides and clindamycin against *T. pallidum* in syphilis cases [159].

Overall, due to the proposed oral origin of AD and neurodegeneration, the primary preventative action would be to support and provide early treatment for patients with periodontitis in order to reduce the incidence and progression of unstable deep pockets and to provide and encourage an efficient oral hygiene regime to avoid the bacterial biofilm build-up. This would reduce peripheral infection load and the subsequent systemic inflammation. Thus, implementing effective preventative strategies against periodontitis may be a relatively straightforward means of reducing neurodegenerative disease processes and/or delaying their onset.

## Figures and Tables

**Figure 1 ijerph-20-06049-f001:**
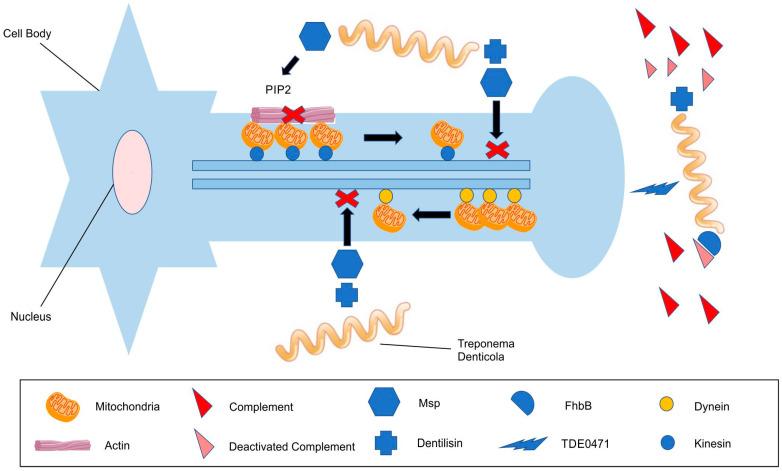
*Treponema denticola* and its effects on the peripheral axon: cytoskeletal impairment, axonal transport disruption with mitochondrial docking, and complement evasion.

## Data Availability

The data presented in this study are openly available on Pubmed at https://pubmed.ncbi.nlm.nih.gov.

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
