# Peer review of "Treponema denticola Has the Potential to Cause Neurodegeneration in the Midbrain via the Periodontal Route of Infection—Narrative Review"

_ijerph, 2023, doi:10.3390/ijerph20116049_

Round 1
Reviewer 1 Report
In this review, Pisani et al. described the potential of Treponema denticola cause neurodegeneration in the midbrain via the periodontal route of infection. While the overall topic would be of interest to the readers of International Journal of Environmental Research and Public Health, the current version will require major changes in order to improve the overall quality of this manuscript. Please see the major issues below.
1. Abstract:
(1) This section is written in a poor manner. There are a lot of grammar mistakes. For instance, it is hard to understand the sentence from line 25 to 28. It should be written in concise sentence.
(2) line 32: cytoskeletal dysregulation is stated abruptly. Please make it clearer what cause cytoskeletal dysregulation?
2. Introduction: In paragraph two, the explanation and the association of amyloid-beta (Aβ) plaques, neurofibrillary tangles(NFTs) and phosphorylated tau(p-tau) should be added. Also, their roles in Alzheimer’s disease should be added.
3. It is difficult to understand the subtitle Evading the Complement Cascade, The Cytoskeletal impairment, Axonal transport disruption can cause neuronal apoptosis and Docking of mitochondrial trafficking due to axonal transport impairment. These four parts can be constructed in one section with the subtitle The mechanism of neurodegeneration by T. denticola. Besides, the part of Axonal transport disruption and Docking of mitochondrial trafficking should be written in the part of the cytoskeletal impairment as they are the consequences of the cytoskeletal impairment.
4. It should be clarified which part means the cytoskeletal impairment in Figure 1.
5. Line378: the research papers should be cited.
Author Response
Reviewer 1
- Abstract: 1. done. thanks. 2. done, thanks.
- Intro: thanks for the objection, but we do not think that represents a something absolutley relevant in the context of this review as not really focused on this matter.
- all unified inthe same paragraph, although divided by sections. many thanks. done it.
- the figure actually provides a good understanding about which parts of the cytoskeleton are involved and impaired. the legenda provides a good explanation of the different subcomponents.
- research paper cited. thanks a lot for highlighting this!
Reviewer 2 Report
This article discussed the relationship between AD and periodontal disease and proposed a potential bacterial mechanisms that could damage the NE neurons and lead to the neurotransmitter deficit. There are two points should be improved.
1> Some relevant clinical charts of the bacteria will cause periodontal disease should be added to make the results more reliable.
2>Please give evidence for a more direct bacterial role in causing AD and rule out other possible factors.
Author Response
Reviewer 2
many thanks for your advices.
- Other perio bacteria are mentioned in the review as Pg, however this paper is focusing on T denticola as the actual trigger of the induced neuropathy. thanks.
- It is actually mentioned in all the sections. The central effects are clearly highlighting the direct role of T denticola in the tissue damage and its clear resistance via the biofilm aggregation. Many thanks for the advice.